# Electronically phase separated nano-network in antiferromagnetic insulating LaMnO₃/PrMnO₃/CaMnO₃ tricolor superlattice

Qiang Li [1,2,9], Tian Miao[2,3,9], Huimin Zhang[2,4,5,9], Weiyan Lin[1], Wenhao He[1,2], Yang Zhong[2,4,5], Lifen Xiang[2], Lina Deng[2], Biying Ye[2], Qian Shi[2], Yinyan Zhu[1,5,6], Hangwen Guo[1,5,6], Wenbin Wang[1,5,6], Changlin Zheng[1,2], Lifeng Yin[1,2,5,6,7,8], Xiaodong Zhou [1,5,6] ✉, Hongjun Xiang [2,4,5] ✉ & Jian Shen [1,2,5,6,7,8] ✉

Strongly correlated materials often exhibit an electronic phase separation (EPS) phenomena whose domain pattern is random in nature. The ability to control the spatial arrangement of the electronic phases at microscopic scales is highly desirable for tailoring their macroscopic properties and/or designing novel electronic devices. Here we report the formation of EPS nanoscale network in a mono-atomically stacked LaMnO₃/CaMnO₃/PrMnO₃ superlattice grown on SrTiO₃ (STO) (001) substrate, which is known to have an antiferromagnetic (AFM) insulating ground state. The EPS nano-network is a consequence of an internal strain relaxation triggered by the structural domain formation of the underlying STO substrate at low temperatures. The same nanoscale network pattern can be reproduced upon temperature cycling allowing us to employ different local imaging techniques to directly compare the magnetic and transport state of a single EPS domain. Our results confirm the one-to-one correspondence between ferromagnetic (AFM) to metallic (insulating) state in manganite. It also represents a significant step in a paradigm shift from passively characterizing EPS in strongly correlated systems to actively engaging in its manipulation.

Strongly correlated systems are well known for their strong tendency to form multiple electronic phases simultaneously in real space due to the interplay of charge, spin, and orbital degrees of freedom[1]. If the spatial arrangement of these phases can be manipulated, one may build novel electronic and spintronic devices in one single material with the advantage of not having chemical interfaces that usually degrade the charge and spin transport efficiency[2]. Achieving such designer's pattern of electronic domains in one single material requires the competing electronic phases to have close energy levels yet with a high energy barrier for preventing phase diffusion or intermixing. In addition, the constituent phases should retain their high susceptibility to external stimuli, which makes writing and erasing such phases possible in a functional device. Successful attempts include patterning electronic

[1]State Key Laboratory of Surface Physics and Institute for Nanoelectronic Devices and Quantum Computing, Fudan University, Shanghai 200433, China. [2]Department of Physics, Fudan University, Shanghai 200433, China. [3]School of Materials Science and Engineering, Xi'an Jiaotong University, Xi'an, Shanxi 710049, China. [4]Key Laboratory of Computational Physical Sciences (Ministry of Education) and Institute of Computational Physical Sciences, Fudan University, Shanghai 200433, China. [5]Shanghai Qi Zhi Institute, Shanghai 200232, China. [6]Zhangjiang Fudan International Innovation Center, Fudan University, Shanghai 201210, China. [7]Shanghai Research Center for Quantum Sciences, Shanghai 201315, China. [8]Collaborative Innovation Center of Advanced Microstructures, Nanjing 210093, China. [9]These authors contributed equally: Qiang Li, Tian Miao, Huimin Zhang. ✉e-mail: zhouxd@fudan.edu.cn; hxiang@fudan.edu.cn; shenj5494@fudan.edu.cn

phases in LaAlO$_3$/SrTiO$_3$ two-dimensional electron gas using atomic force microscope tip[3–5] and in (La$_{1-y}$Pr$_y$)$_{1-x}$Ca$_x$MnO$_3$ (LPCMO) manganite thin films using edge effects[6–8]. The stability of the patterned phases, however, rely on local stoichiometry changes upon the application of a local field.

The recently discovered LaMnO$_3$(LMO)/PrMnO$_3$(PMO)/CaMnO$_3$(CMO) tricolor superlattice offers an ideal platform to achieve such designer's pattern of electronic phases[9]. The superlattice has a robust antiferromagnetic (AFM) insulating ground state and requires a high magnetic field (>30 T) to be transited into a ferromagnetic (FM) metallic state. On the other hand, the superlattice can be viewed as a chemically ordered LPCMO manganite, which can be easily turned into an electronically phase separated (EPS) state consisting of AFM insulating and FM metallic phases by introducing slight disorders via randomizing A-site cations or a non-uniform lattice distortion[10,11]. The non-uniform lattice distortion can be conveniently achieved by application of a local strain field, which allows the formation of local new phases in the AFM insulating matrix[12,13]. The new phases should have a well-defined boundary because the phase diffusion is naturally prohibited by the surrounding tricolor superlattice structure with different strain states.

In this work, we demonstrate the formation of an EPS nanoscale network in the AFM insulating matrix in the LMO/PMO/CMO tricolor superlattice grown on SrTiO$_3$ (STO) (001) substrate. This nanoscale network is a consequence of an internal strain relaxation triggered at the structural domain wall (DW) of the underlying STO substrate at low temperatures. The nano-network of the new phase is reproducible upon temperate cycling, which allows us to study its magnetic and transport states by directly comparing the magnetic force microscopy (MFM) and scanning microwave impedance microscopy (sMIM) images acquired from the same location. Our results directly confirm that the FM (AFM) states of a EPS domain is also metallic (insulating) in manganite. For comparison, the tricolor superlattice grown on NdGaO$_3$ substrate, which has no structural DW, remains as a uniform AFM insulating state throughout all temperature and field ranges (see supplementary SI E)[9].

## Results

### Structure characterizations of nanoscale network of EPS domain

Figure 1a shows the schematic of our experimental set-up, including MFM and sMIM for characterizing local magnetic and transport states, respectively. The 60 nm mono-atomically stacked LMO/PMO/CMO tricolor superlattice was grown on STO (001) substrate, which can be viewed as a fully A-site chemically ordered LPCMO system with a stable single AFM insulating phase as its ground state[9]. We have conducted a careful X-ray diffraction (XRD) measurement to confirm the epitaxial growth of the tricolor superlattice film on STO substrate, i.e., the film is in a tensile strained state with the same lattice constant of that of the substrate (see supplementary SI A). Interestingly, after one cooling-warming temperature cycle, a nanoscale network emerges in the sample, as shown by the atomic force microscope image in Fig. 1b. The network is formed by morphologically protruded ridges with a height less than 1 nm as inferred from an averaged line-cut height profile at the bottom of Fig. 1b. These nano ridges are aligned along [100], [010], [110]

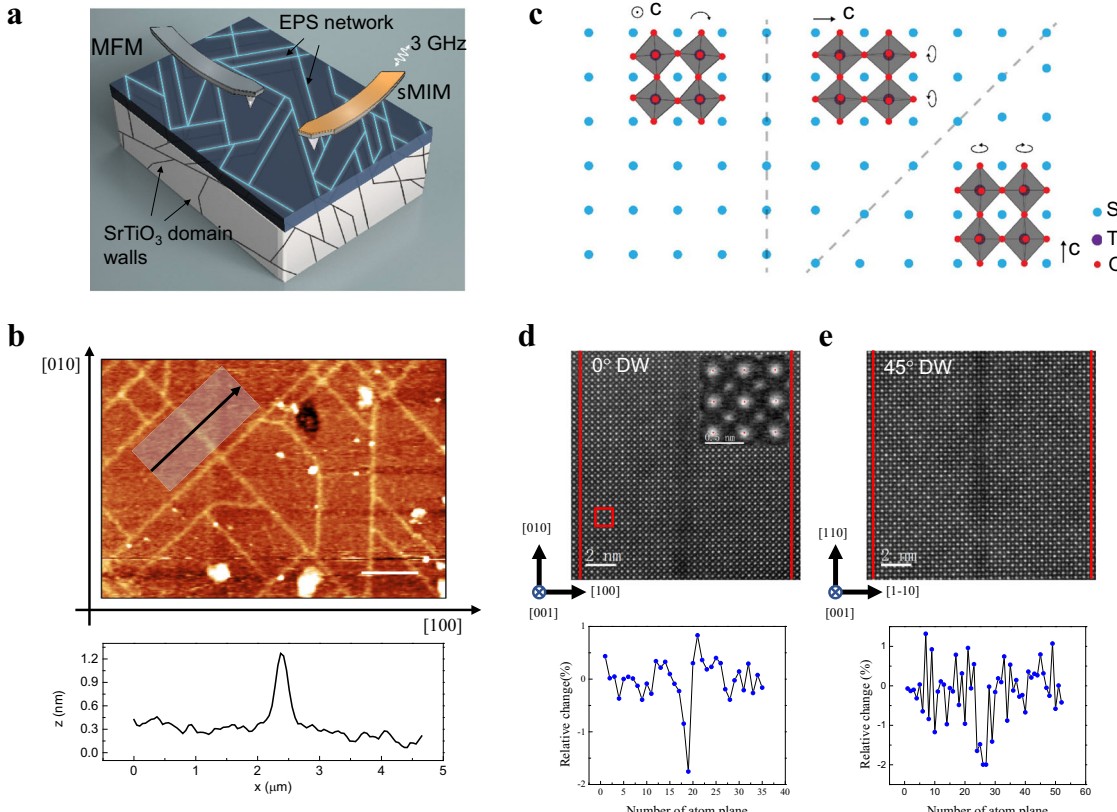

**Fig. 1 | Structure characterization of nanoscale network of electronic phase separated (EPS) domain. a** The schematic of experimental set-up. A nanoscale network of structure distortion exists in LaMnO$_3$/PrMnO$_3$/CaMnO$_3$ tricolor superlattice due to the formation of twin boundaries in SrTiO$_3$ (STO) substrate. Magnetic force microscopy (MFM) and scanning microwave impedance microscopy (sMIM) were adopted to characterize the such network. **b** A typical atomic force microscope topographic image shows the such nanoscale network of structure distortion with a representative line-cut height profile at the bottom. The scale bar is 2 μm. **c** The schematic of twin domains and domain wall (DW) boundaries of STO in a tetragonal phase. Two types of DW boundary exist, extending along 0° and 45°. **d, e** Scanning transmission electron microscopy characterization and strain analysis of such 0° and 45° DW. The scale bar is 2 nm.

and [−110] directions, indicating a close connection with the structural transition of the underlying STO (001) substrate. The structural transition of the STO substrate occurs with a rotation of the TiO$_6$ octahedron around the lengthened axis (c axis) at 105 K from a high-temperature cubic phase to a low-temperature tetragonal phase[14–16]. Elongated domains along all three original cubic axes form together with twin boundaries (Fig. 1c). These twin boundaries are categorized into two types when projected onto the (001) plane. If the c axis of the two domains lies in the (001) plane, the 45° and 135° twin domain boundaries form. If the c axis of one domain is out-of-plane, the 0° and 90° twin domain boundaries form. We refer the former (latter) to 45°(0°) DW hereafter to simplify data description. From Fig. 1b and c, it is very tempting to associate the DW structure of the STO substrate to the nano-network seen in LMO/PMO/CMO tricolor superlattice, as shown in Fig. 1a. Considering the fact that the tricolor superlattice is highly strained because of both the lattice mismatch between the component materials and the tensile strain exerted by the substrate, the formation of the nano ridges is likely the consequence of an internal strain relaxation of the tricolor superlattice triggered by the STO DW. We show additional evidence of structural dislocations at the nano-network regions as the mechanism to release the strain (see supplementary SI B). Note that the nano ridges are absent in the as-grown state and will only appear after one cooling-warming cycle (see supplementary SI A), which further links it to the STO structural transition at low temperatures. Interestingly, although STO DW will disappear at high temperatures, the nano ridges left over in the superlattice thin film is persistent up to room temperature. Moreover, the nano-network pattern in the superlattice doesn't change anymore despite of new DWs appearing randomly on STO substrate in subsequent thermal cycling. This implies that most of the internal strains are released in the first cool-down, after which the film is in a strain-relieved and structurally stable state without the need for further structural relaxation.

We perform aberration-corrected scanning transmission electron microscopy (STEM) to further characterize the structure and chemistry of the tricolor superlattice and in particular, the nano ridges (see supplementary SI B for more STEM results)[13]. Figure 1d and e show the in-plane high-angle annular dark-field scanning transmission electron microscopy (HAADF-STEM) image of a typical 0° and 45° DW, respectively. The DW contrast can be clearly seen in the image, and the DW area maintains the atomic registry of the original lattice. Note that the width of such DW determined in STEM imaging is less than 2 nm, which is much smaller than that seen in Fig. 1b (~40 nm) because the latter is due to the limited spatial resolution of the particular atomic force microscope setup. A strain analysis of such an image was performed using a real-space atom position finding analysis[17]. The red dots in the inserted image of Fig. 1d show the located centers of the atoms at A site. The line profile at the bottom of Fig. 1d shows the relative change of the averaged lattice spacing along the [100] direction when crossing the 0° DW. It clearly shows a 1.8% reduced lattice spacing, or a compressive strain, as compared to that of the domains at two sides. A compressive strain could also be found in the 45° DW with a similar amplitude (-1.9%). In addition, such a compressive strain is uniaxial in nature (see supplementary SI B). The STEM data shown here supports our speculation that the epitaxial tensile strain is relieved at the DW.

## MFM characterizations of nanoscale network of EPS domain

This spatial variation of the strain state in the film turns out to have a dramatic effect on its electronic and magnetic properties. Figure 2 shows a series of field-dependent MFM images acquired from the same sample location shown in Fig. 1b after a zero field cooling to 12 K. The applied magnetic field is labeled at the top right-hand corner of each MFM image. These images reveal rich magnetic features: 1) All areas except the nanoscale network regions exhibit a zero-phase signal in MFM images, which is consistent with the previous reports that the tricolor superlattice has a single uniform AFM insulating ground state[9]. The global transport measurement confirms this highly insulating state (see supplementary SI A). The AFM state is rather robust that cannot be transited into FM state under high external magnetic field (>30 T), explaining why no magnetic field evolution was observed in these areas; 2) The 0° DWs (horizontal and vertical lines in the images) are in a uniform FM state. When the external field is small (Fig. 2a, b), the magnetization of such 0° DWs lies in the plane leading to weak MFM

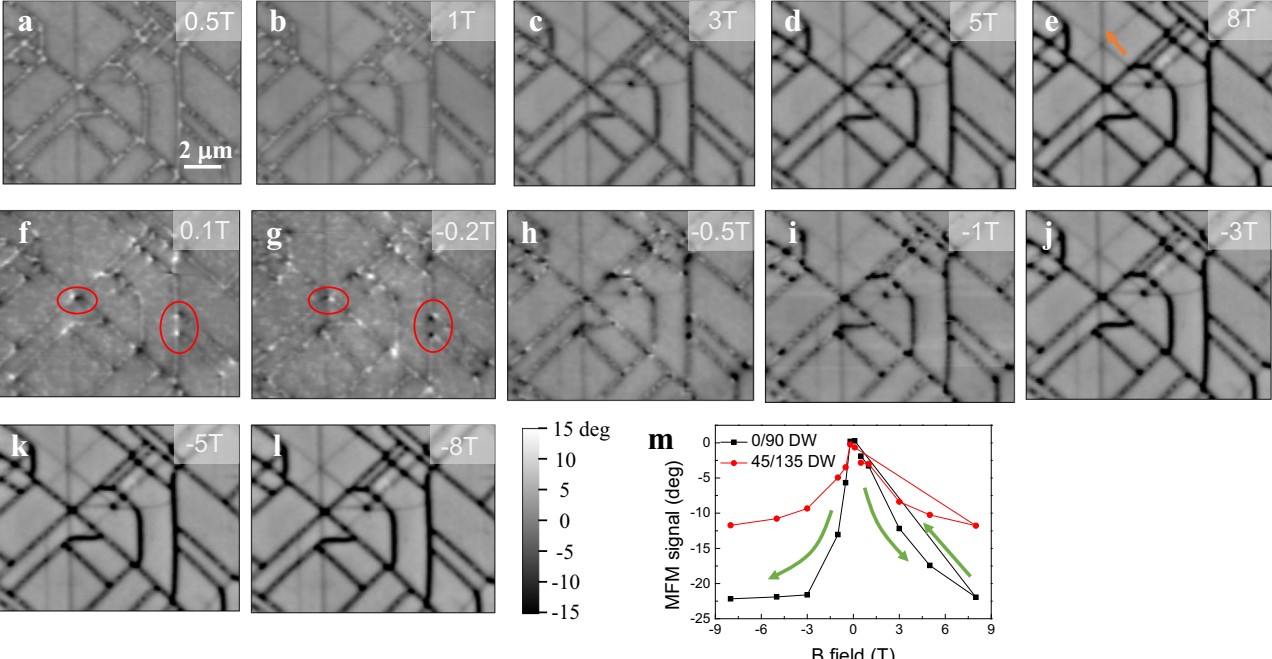

**Fig. 2 | Magnetic force microscopy (MFM) characterization of nanoscale network of electronic phase separated (EPS) domain. a–l** Sequential MFM images taken at 12 K under different magnetic fields. Ferromagnetic metallic state emerges from an otherwise stable single antiferromagnetic insulating state along a nanoscale network forming EPS domain. The scale bar is 2 µm. **m** Averaged MFM signal levels of the nanoscale network as a function of magnetic field.

signals indistinguishable from the background AFM state. With increasing field (Fig. 2c–e), the magnetization rotates from in-plane to out-of-plane giving rise to a negative-phase signal (black) with increasing amplitude. As the field is reduced to zero, the magnetization rotates back to in-plane (Fig. 2f–h). It points out-of-plane again under large negative fields (Fig. 2i–l). The in-plane magnetic anisotropy of such FM state in 0° DWs can also be unveiled from the behavior of the crossing points denoted in Fig. 2f, g. This crossing point is the end of FM state with an in-plane magnetization, whose magnetic flux profile thus looks like an in-plane dipole. This gives rise to a white-black dumbbell pattern in the MFM image, which flips as the external field changes from 0.1 T to −0.2 T due to the flip of the tip's magnetization; 3) The 45° DWs (diagonal lines in the images) consist of multiple segments with black and white contrast corresponding to the FM and AFM phase (Fig. 2i), respectively, indicating that they are in a mixed phase state. Parts of AFM phase are transited into FM phase when the external field increases (compare Fig. 2i to Fig. 2l); 4) There are some gray lines as denoted by the orange arrow in Fig. 2e. These lines are absent in the corresponding atomic force microscope image (Fig. 1b), which may reflect structural distortion well underneath the top surface whose FM stray field gets attenuated towards the surface resulting in a smaller MFM signal. Figure 2m shows the averaged MFM signal levels of the 0° and the 45° DWs as a function of the magnetic field, confirming our observation that the 0° DW has a higher FM phase volume than that of the 45° DW.

What is mostly striking in this MFM characterization is that EPS domains emerge from such a robust AFM background to form a nanoscale network. This kind of EPS pattern is totally different from natural EPS observed in manganite, featuring domains with irregular shapes and intertwined configurations[10,18,19]. The nanoscale network arises from a local response to the formation of the structural DWs in STO substrate at low temperatures. Once formed, the network of EPS domains would reappear at the same locations after each round of temperature cycle. It appears that the crystallographic orientation has an influence on the domain distribution as well, i.e., a uniform FM phase exists in the 0° DWs while it coexists with AFM phase in the 45° DWs. Also note that the AFM phase in 45° DWs is less stable against the magnetic field than that of distortion-free areas.

## sMIM characterizations of nanoscale network of EPS domain

The fact that the EPS network maintains its location allows us to directly compare the magnetic and transport states of a single EPS domain by employing different imaging techniques that are sensitive to magnetic and transport properties, respectively. In addition to MFM characterization of the magnetic state, we employ sMIM to probe the transport state of the same EPS domain. sMIM is a recently developed scanning probe technique capable of local conductivity imaging (see supplementary SI C for the detailed introduction)[20]. In sMIM, a 3 GHz microwave is delivered through the atomic force microscope tip to interact with the sample area underneath the probe (Fig. 1a). The reflected microwave signal is collected, which measures the screening property of the local sample area. Different from MFM, sMIM characterizes the transport state (local conductivity) of a EPS domain[21–23].

Figure 3a–c show field-dependent sMIM images acquired from the same area of Fig. 1b in which the higher sMIM signal represents a higher local conductivity. Figure 3d shows the averaged sMIM signal levels of the 0° and the 45° DWs as a function of the magnetic field. The error bar shown in Fig. 3d reflects the spatial variation of sMIM signal level (conductivity) among the DWs. Without any presumptions, we draw conclusions directly from the sMIM images that the nanoscale network is more conductive than the rest distortion-free areas and the 0° DWs, on average, have a higher conductivity than that of the 45° DWs. Comparing with the MFM images acquired from the same EPS network, we can safely claim that in the EPS network there is a one-to-one correspondence between the FM (AFM) state to the metallic (insulating) state. The higher conductivity in the 0° DWs reflects their more uniform ferromagnetism, i.e., it has a larger volume fraction of FM metallic phase than that of the 45° DWs as explicitly showcased in MFM images of Fig. 2. The enhanced conductivity for both the 45° and the 0° DWs with increasing field results from the AFM to FM phase transition in the 45° DWs and the magnetoresistance behavior in the 0° DWs.

The one-to-one correspondence between the magnetic and transport state of a single EPS domain is essential for resolving a puzzle in a manganite., i.e., the metal-insulator transition (MIT) temperature derived from a global resistance measurement is often not the same as

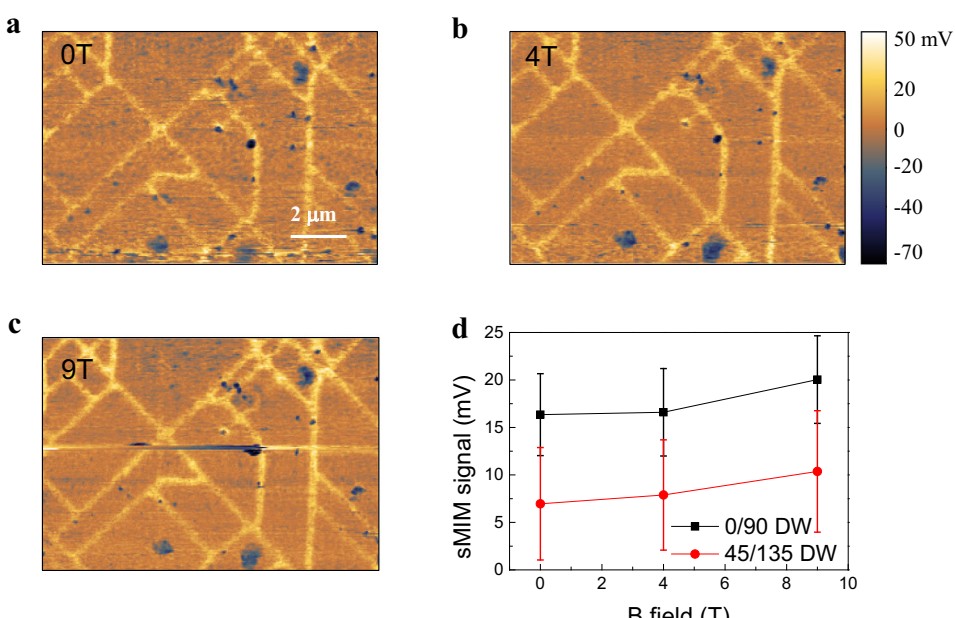

**Fig. 3 | Scanning microwave impedance microscopy (sMIM) characterization of nanoscale network of electronic phase separated domain. a–c** sMIM images taken at 2 K under different magnetic fields. Ferromagnetic metallic state with enhanced conductivity is resolved along such nanoscale network. The scale bar is 2 μm. **d** Averaged sMIM signal levels of the nanoscale network of domain wall (DW) as a function of magnetic field.

the magnetic Curie temperature determined from the global magnetization measurement[10]. Only after the correspondence between FM (AFM) state and metallic (insulating) state is firmly established at a single EPS domain level is this discrepancy in global measurements resolved. It indicates that the MIT in manganite is percolative in nature and the percolation transition in transport (MIT temperature) will happen later than the initial domain formation (Curie temperature). Note that the same correspondence has been observed in another multi-probe imaging work on $La_{2/3}Ca_{1/3}MnO_3$ manganite using simultaneous imaging of MFM and scattering-type scanning near-field optical microscopy[24].

The sMIM imaging also justifies our assignment of a uniform FM metallic state to the 0° DWs area. In MFM characterization of Fig. 2, one obtains a zero-phase at low fields in 0° DWs, which changes to a large negative-phase at high fields. Since MFM only detects the out-of-plane magnetic flux, it cannot differentiate a FM phase with an in-plane magnetization from an AFM phase. Both scenarios give rise to the same field evolution seen in Fig. 2. However, the FM and AFM phases differ in their conductivity and sMIM imaging at 0 T in Fig. 3a rules out the possible existence of the AFM phase in the 0° DWs as it shows a higher conductivity than the surroundings.

## Density functional theory calculations

Now we want to understand why FM metallic phase emerges here from an otherwise very stable AFM insulating phase in LMO/PMO/CMO tricolor superlattice. The STEM measurement gives us a strong hint that the local strain field drives such a nanoscale network of EPS domain. To examine the role of the strain, we perform ab initio density functional theory (DFT) calculations to determine the magnetic ground state and its dependence on the lattice parameters. We first consider the lattice structure of such LMO/PMO/CMO tricolor superlattice. In the perovskite structure with a formula $ABX_3$, the rotation and tilting of $BX_6$ octahedron happen occasionally, such as in manganite, when there is a mismatch between the ionic radius of A-site and the cubo-octahedral cavity. The Glazer notation was developed to describe the octahedral tilting distortions[25]. For bulk materials $LaMnO_3$, $PrMnO_3$, and $CaMnO_3$, all form the $a^-a^-c^+$ octahedral tilting mode. As in the LMO/PMO/CMO tricolor superlattice, several structures with different tilting modes and different symmetry are considered. Our results show that the $a^-a^-c^+$ tilting mode is the most stable one (see Table S1 in the supplementary SI F). Figure 4a shows the lattice structure of such A-site ordered tricolor superlattice with an $a^-a^-c^+$ tilting mode as its structural ground state. We calculate the energies of several different magnetic states as

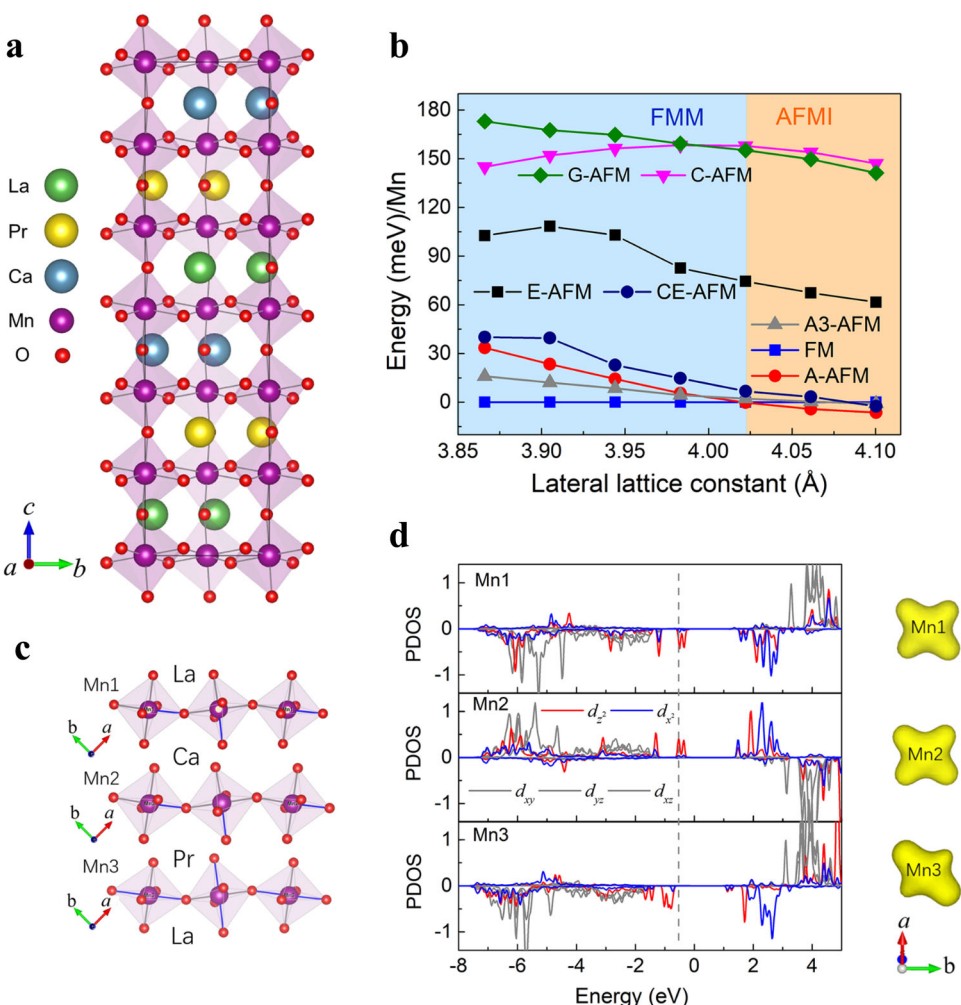

**Fig. 4 | Density functional theory calculations of electronic states in LaMnO₃/ PrMnO₃/CaMnO₃ tricolor superlattice. a** The A-site ordered tricolor superlattice structure. **b** The energy of different magnetic states varying with different SrTiO₃ lattice parameters. FMM (AFMI) stands for ferromagnetic metallic (antiferromagnetic insulating) phase. The energy of FM state is set as a reference for the energy comparison. **c** The three different Mn-O octahedral layers are surrounded

with different A-site ions, which are marked as Mn1, Mn2, and Mn3. The blue Mn-O bonds represent the elongated orientation. **d** The projected density of states (PDOS) of Mn's 3d orbitals for three kinds of Mn ion mentioned above. The PDOS of $t_{2g}$ orbitals are marked in gray color, and the two $e_g$ orbitals are marked in red color for $d_{z^2}$ orbital and blue color for $d_{x^2-y^2}$ orbital. The partial charge density of three different Mn ions are shown on the right of PDOS.

a function of lattice parameters of STO substrate (Fig. 4b). We take the cubic phase of STO in which $a$ and $b$ axes are equal and let lattice parameters of tricolor superlattice thin film to be equal to STO as a result of the epitaxial growth. In Fig. 4b, among various magnetic states, the A-type AFM is the most stable state in the tensile strain range corresponding to larger lattice parameters on the right side. As this lattice parameter gets shorter towards the compressive strain, the FM state becomes the magnetic ground state. The phase transition under the biaxial strain can be apparently seen in Fig. 4b. We also check the uniaxial strain case which shows the same phenomenon (see supplementary SI F). In brief, the magnetic phase of LPCMO tricolor superlattice should transit from AFM to FM state when the lattice parameter of STO becomes smaller in agreement with our experimental observations. Note that, what matters in this calculation is the global trend of phase transition from AFM to FM state with reducing lattice parameters rather than an absolute lattice value (or strain state) denoted here because the DFT-predicted lattice constant often deviates much from actual experimental conditions. Regarding the actual strain state, the nanoscale network of FM metallic phase is more likely in a tensile strain-relieved state compared to domains on two sides. Similar hidden FM metallic state suppressed by the tensile strain has been reported in other manganite systems[24].

We also carefully study the electronic property of Mn ions in this special LMO/PMO/CMO tricolor superlattice. We classify the Mn ions in different layers to three types marked as Mn1, Mn2, and Mn3, according to their distinct nearby A-site cations (Fig. 4c). The Mn-O bonds marked in blue represent the elongated ones of six Mn-O bonds in an octahedron. The Mn3-O octahedron is more likely to form the Jahn-Teller distortion with two elongated bonds and four shortened bonds referring to a regular octahedron. However, in the Mn1-O and Mn2-O octahedrons, only one elongated bond is found possibly due to the smaller radius of $Ca^{2+}$ than that of $La^{3+}$ and $Pr^{3+}$. The difference between Mn3 and Mn1(Mn2) is also reflected in its projected density of state (PDOS) in A-AFM state (Fig. 4d). The corresponding partial charge density of Mn ions is shown on the right of PDOS. The Mn3 ion is shaped like the $d_{z2}$ orbital, which is in accordance with the PDOS that one $e_g$ electron occupies the $d_{z2}$ orbital. With respect to Mn1 and Mn2, both shapes of its occupied orbitals are more like a mixed $d_{z2}$ and $d_{x2-y2}$ orbital, since the PDOS reveals a hybridization of $d_{z2}$ and $d_{x2-y2}$. As discussed above, in the A-type AFM state, the Jahn-Teller distortion happens in Mn3-O octahedron which splits $d_{z2}$ orbital from $d_{x2-y2}$ orbital rendering Mn3 ion +3 valence state. On the other hand, Mn1 and Mn2 have a mixed valence. The band splitting and gap opening in A-type AFM may be due to the hybridization between $d$ orbitals of Mn1 and Mn2, leading to the fact that one electron occupies the bonding orbital and leaves the antibonding orbital empty. When the magnetic state transits from AFM to FM state, our calculation indicates that the six Mn-O bonds in all of those octahedrons get shorter and become closer, and the elongated bonds will not be obvious (see supplementary SI F). Accordingly, the Jahn-Teller distortion of Mn3-O octahedron becomes weaker in FM sate, so that the $d_{z2}$ and $d_{x2-y2}$ prefers two-fold degeneracy rather than splitting, thus inducing a transition from an insulating to metallic state. Note that the metallic state usually favors a FM coupling due to a kinetic energy gain in the itinerant state.

## Discussion
While strain is known to drastically influence the global physical behavior of manganite[26–28], our results demonstrate that electronic state can be controlled at the microscopic scale using local strain field. The unique nanoscale network of EPS domain established here in tricolor superlattice film completely differs from conventional EPS found in many other manganite systems with intertwined spatial configurations. This contrasting behavior is better appreciated by comparing it to our previous work in which we grew two kinds of LPCMO thin films on STO

substrate, i.e., one has a random A-site mixing and the other has a partial A-site order with [$La_{0.625}Ca_{0.375}MnO_3$/$Pr_{0.625}Ca_{0.375}MnO_3$] superlattice[10]. Both systems show conventional EPS state with intertwined shape even they experience the same strain field from STO substrate at low temperatures. It is the formation of a fully A-site ordered LPCMO film that gives rise to such a peculiar local strain response and a new type of EPS state in manganite. Such artificial EPS excels in its spatial selectivity and inspires a new paradigm for building nano-electronic and spintronic devices on one single material. Looking ahead, new spatially patterned methods on the substrate are called for to design and exert a local strain field on the same tricolor superlattice film to realize a more controllable EPS domain with writing and erasing at will[29,30]. Taking advantages of extreme sensitivity of electronic states in such tricolor superlattice to local external strain stimuli, this approach will achieve a spatially patterned EPS on demand in analogy to the circuit pattering in traditional silicon-based devices (see supplementary SI G for our preliminary results on the formation of a local FM metallic phase by using tip to mechanically press the same superlattice film).

## Methods
### Sample growth
The 60 nm [(LMO)1/(PMO)1/(CMO)1]$_{52}$ tricolor superlattice was epitaxially grown on $5 \times 5\,mm^2$ single-crystal $SrTiO_3$ (001) substrates by pulsed laser deposition (ultraviolet laser in 248 nm, $2\,J/cm^2$ fluence, 2 Hz) at 820 °C. The oxygen pressure was 8 mTorr, including 8% ozone. In-situ reflection high-energy electron diffraction was used to monitor the unit-cell by unit-cell growth process.

### XRD measurements
The X-ray reciprocal space maps and the XRD measurements in Fig. S1 were collected by using Cu-Kα radiation (1.5418 Å, Bruker AXS D8 Discover). Synchrotron XRD measurements in Fig. S4 were carried out at beamline BL14B1 of Shanghai Synchrotron Radiation Facility at room temperature, using 10 keV X-rays (1.239 Å).

### STEM
The atomic resolution STEM-HAADF images were acquired with a field-emission transmission electron microscope (Themis Z, Thermo fisher Scientific) fitted with double aberration correctors (SCORR probe corrector and CETCOR image corrector, CEOS GmbH). The microscope was operated at 300 kV. For STEM imaging, the semi-convergent angle of the probe forming lens was set to 21.4 mrad, and the semi-collection angles of the annular dark field detector are from 79 mrad to 200 mrad. The HAADF-STEM images were acquired with a fast scanning rate (0.5 µs dwell time) to minimize the effects of scan noise and sample drift. 40 frames were integrated using a rigid-body drift correction algorithm to form the drift-corrected HAADF-STEM images. The strain analysis of the HAADF-STEM images was performed with real-space peak finding method using Atomap software[17]. The locations of the atoms were refined with a two-dimensional Gaussian fitting of the local atomic columns.

### MFM
MFM measurement were performed in a commercial AttoDry 1000 system. Commercial Co/Cr-coated MFM probes were used in the MFM dual-pass mode with a lift height of 100 nm in order to remove the morphology contribution from MFM signals.

### sMIM
sMIM measurements were conducted in a commercial AttoDry 2100 system. A 3 GHz microwave signal was delivered to a micro-fabricated shielded probe commercially available from PrimeNano Inc. The reflected signal was demodulated into two signal channels, sMIM-Im and sMIM-Re. sMIM-Im is used throughout this work as it monotonically increases with the local conductivity.

## DFT calculations

Our density functional theory calculations were performed using the projected augmented wave pseudopotentials implemented in the Vienna ab-initio Simulation Package (VASP)[31,32]. The electron interactions were described using the Perdew-Burke-Ernzerhof function revised for solids (PBEsol)[33], and the generalized gradient approximation plus U (GGA + U) method was adopted to treat the exchange and correlation of electrons[34–36]. The Liechtenstein approach was adopted with $U = 3.5$ eV ($U$ is the Coulomb repulsion interaction) and $J_H = 0.9$ eV ($J_H$ is Hund's coupling) parameters on the 3d electrons of Mn[37]. Our tests show that the qualitative results do not change if other reasonable $U$ values are adopted. The plane-wave energy cutoff was set to be 500 eV. The Monkhorst-Pack k-point mesh is $7 \times 7 \times 2$ for the 60 atoms superlattice and $3 \times 3 \times 2$ for the 180 atoms superlattice corresponding to different magnetic states with PBEsol functional. For obtaining the more accurate electronic structure, the hybrid functional calculations with the Heyd-Scuseria-Ernzerhof (HSE) functional[38–40] have been performed to calculate the electronic density of states (DOS). The Monkhorst-Pack k-mesh is reduced to $2 \times 2 \times 2$ for the 60 atoms superlattice.

## Data availability

All raw and derived data used to support the findings of this work are available from the authors upon reasonable request.

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

## Acknowledgements

The authors acknowledge support from the National Natural Science Foundation of China (Grant No. 12274088, 12074080, 12074071, 61871134, 62171136, 811825403, 11991061, 12188101, 11991062, 12074075 and 11904052). The work was additionally supported by the Shanghai

Science and Technology Committee Rising-Star Program (19QA1401000), Shanghai Municipal Science and Technology Major Project (2019SHZDZX01), Shanghai Municipal Natural Science Foundation (20501130600 and 22ZR1407400), National Key Research Program of China (2016YFA0300702) and the Postdoctoral Science Foundation of China (2021TQ0265).

## Author contributions

J.S., H.X., and X.Z. supervised the research. Q.L. performed the MFM and XRD measurements assisted by L.X. T.M. prepared the samples and conducted global transport and magnetization characterizations. W.L. performed the sMIM measurement. W.H. and C.Z. performed the TEM measurement. H.Z., Y.Z., and H.X. performed the DFT calculations. L.D., B.Y., Q.S., Y.Z., H.G., W.W., and L.Y. assisted in the data analysis. X.Z. and J.S. prepared the manuscript with comments from all authors.

## Competing interests

The authors declare no competing interests.
