## [Peer Review File · Nature Communications]

Electronically phase separated nano-network in
antiferromagnetic insulating $\text{LaMnO}_3/\text{PrMnO}_3/\text{CaMnO}_3$
tricolor superlatticeREVIEWER COMMENTS

Reviewer #1 (Remarks to the Author):

In this manuscript, Li et al. reports a comprehensive study on the electronic phase separation (EPS) in tricolor LaMnO₃/CaMnO₃/PrMnO₃ superlattice grown on SrTiO₃ substrates. The authors obtained the EPS nano-network by thermal cycling the sample through the cubic-tetragonal phase transition in SrTiO₃, and confirmed the structure of such nano ridges by scanning transmission electron microscopy (STEM). Using field-dependent magnetic-force microscopy (MFM), they demonstrated that the EPS network can be easily driven to ferromagnetic (FM) states, in sharp contrast to the robust antiferromagnetic (AFM) bulk. Using scanning microwave impedance microscopy (sMIM), they showed that the EPS network is more conductive than the background domains. Finally, they put forth a density-functional-theory (DFT) based analysis to explain the emergence of FM metallic phase at the nano-network. This work is of high quality. I can recommend its publication in Nature Communications after the authors consider the following comments.

1. The EPS network was formed during the first cooldown of the sample through the structural transition of SrTiO₃ due to lattice strain at the DWs. The authors claimed that the EPS reappears at the same locations after each round of temperature cycle. However, the SrTiO₃ substrate could still form different tetragonal domain structures at different cooldowns. Why wouldn't they observe more networks during subsequent thermal cycles?
2. The MFM data in Fig. 2 clearly demonstrate the FM nature of the EPS network. In contrast, the discussion on sMIM data is problematic. What does the technique measure? How does one relate its signals (in mV from the images) to local conductivity, as claimed by the authors? Such critical information is not found in the text, Methods, or SI.
3. Following point #2, the sMIM data analysis is not convincing. The authors show that (in Fig. 3b) the 0/90 walls exhibit signals twice as high as the 45/135 walls, without giving the error bars. As far as I can tell, the 45/135 DWs look just the same as the 0/90 DWs, especially in the upper-right and upper-left corners of the images. I also don't find any discussions on why there should be difference in conductivity between these two types of walls. In my opinion, unless they have compelling data and good understanding on the conduction mechanism between these two walls, they should just report the observation of metallic states at the EPS network and the (slight but discernible) field dependence within some experimental errors.
4. Fig. 3c is rather out of place. It should go with Fig. 2 instead.

Reviewer #2 (Remarks to the Author):

In this manuscript, Li and coauthors present results on a study of manganite superlattices (SLs) (so-called "tri-color superlattices – with lanthanum manganite, praseodymium manganite, and calcium manganite) fabricated on SrTiO₃ (STO) substrates. When the samples are cooled below 105 K and then heated back up, a peculiar domain structure appears in the system, which is apparently caused by the formation of structural twins in the STO substrate below its cubic – tetragonal phase transition which occurs at 105 K. What is rather interesting is that the domain structure then persists in the SL, such that the pattern created is visible by scanning probe microscopy techniques, in both the surface topography, and the local conductivity and magnetic properties. The authors suggest that this is a form of 'electronic phase separation' in the manganite layers induced by the local strain from the substrate when it passes the structural phase transition. Various types of domain walls are observed by scanning transmission electron microscopy (STEM), where a significant local strain is identified. Density functional theory (DFT) of the manganite system is then considered to try to understand the influence of the local strain in the domain walls and thus why a local ferromagnetic conductive phase

appears within the nominally insulating antiferromagnetic matrix of the system.

The results are interesting, the sample quality appears to be high, and the paper is well presented (aside from some issues with figure presentation; see below) and written carefully with quite good flow. Although I do not recommend publication of this manuscript in the current form, if the authors are able to strongly reply to the below comments and concerns, I feel the paper would be worthy of consideration for publication in Nature Communications.

Local strain field and domain wall formation mechanism

On line 69, the authors write, "This nanoscale network strictly follows the structural domain wall (DW) pattern of the underlying STO substrate which exerts a local strain field on the LMO/PMO/CMO superlattice." But the authors do not show evidence for this local strain field. It would be nice to see direct evidence of the patterns formed in the STO substrate – is there a way to see this in real space on a bare substrate using AFM at low temperature?

Why do the authors run the MFM at 12 K and then the transport measurements by sMIM at 2 K? Wouldn't it be better to do direct comparisons at the same temperatures? Also, why is conductive tip AFM not used to characterize the local transport? It could be a much easier way to also see the conductivity of these local domain walls.

What is the mechanism for the domain wall formation and then persistence (even above 105 K when STO becomes cubic again)? It is surprising that the domain structures would appear in the exact same location after several temperature cycles. One would expect that in STO the domains would simply appear at random locations, and these would not be repeatable. More importantly, why do the domains persist in the film layer itself? This is an important point to elucidate because without this we cannot be sure that the effects are not from some localized defects in the samples. Can we see the STEM of the samples near the substrate? Are there any misfit dislocations?

Another related question is to do with the repeatability of this experiment. It would be nice to include in the supplementary some other AFM and MFM images of other samples where this phenomenon is observed but the domain structure is different. In other samples, is the same repeatability of the spatial locations of the domain under repeated temperature cycling also shown?

On a similar note, it would be very interesting to see that if these SLs are grown on a different substrate (e.g. LSAT) which doesn't exhibit the structural phase transition at 105 K, then upon temperature cycling do these domain structures NOT appear?

Structural and chemical characterization of the samples

More structural and chemical characterization of the superlattices is required. I'm particularly interested in knowing how perfect the claimed single unit cell for each of the three components. Some EDS mapping using STEM here would be very helpful. The RHEED data in Fig S1 does assist but it should be better presented and explained to convince the reader that the samples are indeed what the authors say they are.

EDS mapping should also be used on the domain walls – what is the chemistry there? Is it still the same as the matrix surrounding or are they a different phase locally?

Further structural characterization regarding the average strain state of the films is also required. XRD reciprocal space mapping (RSM) would be nice to confirm that the films are tensile strained on the STO substrate and would be a good way to cross check the results of local mapping using the STEM. It would also be nice to compare the magnitude of the strain change required to induce the transition between AFM and FM states.

Figures and presentation of the data

A small but important point. This is less critical for the main figures as they should be copyedited before publication, but for the supplementary information, the text size in the figures is much too small to be legible. This is particularly problematic for Fig. S1 b,d, and Fig. S2. But even better would be to make font size uniform throughout the figures to improve readability. This latter point applies to all the figures, including in the main text. An extreme case is the labels in Figure 1a. These are not even readable. The authors must go through all figures and carefully check the font type (pls make all uniform, e.g. Arial) and sizes.

Figure S1 is too small to read, especially the RHEED and XRD data. Please either make larger or reconsider the layout (could separate into two figures). Also, please index the XRD peaks in Figure S1d.

Figure 4a – please include a bigger schematic and put the colors of the various atoms.

More exciting experiments and work towards applications

It would be highly desirable to show manipulation of the nanoscale EPS network, either by applied force through the AFM tip, or maybe some kind of substrate manipulation (pre-etching and/or in-situ bending?). This kind of manipulation would significantly increase the impact of this paper. I encourage the authors to consider this, and if possible, include such data (even if preliminary) in the supporting information.

Others

The authors should include some other references regarding changes in chemistry and structure at domain walls. One reference that is relevant is Farokhipoor et al., “Artificial chemical and magnetic structure at the domain walls of an epitaxial oxide,” Nature 2014.

In the abstract (line 28), the authors write “controlled formation of EPS nanoscale network...” – this is probably an oversell since the network appears to be formed in a random nature through stochastic processes during the STO structural phase transition. I suggest the authors tone this down.

Reviewer #3 (Remarks to the Author):

This paper reports on nanoscale electronic phase separation (EPS) that emerges at the twin domain boundaries formed in epitaxial thin films of perovskite manganites. The presence of EPS was directly identified by employing local probes for magnetic (MFM) and conducting (MIM) states. Based on the results of these real-space observations, the authors have concluded that the strain relaxation at the domain boundaries induces the FM-metallic state under magnetic fields while the other strained regions are kept to the A-type AFM-insulating state. The conclusion is reasonable and supported well by experimental results. In particular, there are very few reports of direct observations of local EPS by both magnetic and conductive probes at the same positions, which are worth publishing somewhere. However, many papers have already reported the dramatic change in the electronic state via epitaxial strain in manganite thin films. It is also known that biaxial tensile strain tends to stabilize the A-type AFM state, as reported by Z. Fang et al., Phys. Rev. Lett. 84, 3169 (2000). There are also many papers reporting the real-space observations of nanoscale EPS in manganite films employing various local probes. Even the observation of EPS induced by local strain has already been reported in A. S. McLeod et al., Nat. Mater. 19, 397 (2020). What is distinctly new about this study compared to previous papers is the finding of EPS developing only at the twin domain boundaries. However, the domain boundaries are unintentionally introduced associated with the structural phase transition in the

substrates, and hence, their positions are uncontrollable. The origin of the FM-metallic state at the domain boundaries is the epitaxial strain relaxation, which is a known mechanism. Therefore, I feel that the present work lacks exciting novelty to provide substantial advances in this research field and cannot recommend the publication in Nature Communications.

The followings are comments on more detailed points.

1. It is still unclear why the tricolor superlattice is necessary for this work. The sample is nominally a superlattice consisting of mono-atomic layers of three compounds, but the electronic structure will be almost the same as that of the mixed compound in such a short-period superlattice. Even the chemical structure is seemingly very close to the mixed one, judging from the absence of the contrast along the stacking direction associated with the difference in the A-site elements in the cross-sectional STEM images shown in Figs. 1(d) and 1(e), probably due to the chemical diffusion and interface roughness. It is necessary to explain in more detail the advantage of the superlattice structure in this work.

2. The lattice constant of STO is generally 3.905 Å at room temperature. However, the in-plane lattice constant of the film in the strained region evaluated from the STEM image (Fig. 1(d)) is much larger (about 3.96 Å). Why is the in-plane lattice constant of the film so large? Moreover, no results have been shown to prove that the in-plane lattice of the thin film is locked to the substrate.

3. The authors attribute the reduction of the resistance under magnetic fields shown in Fig. S2(a) to the metallic phase at domain boundaries. More detailed consideration will be necessary about whether the change in the conduction at the domain boundaries can explain the observed magnetoresistance.

In this letter we provide a point-to-point response to the reviewers' comments.

In the following, the reviewer's original comments are shown by blue italic characters.

The authors' responses are shown by black normal characters.

Reviewer #1 (Remarks to the Author):

In this manuscript, Li et al. reports a comprehensive study on the electronic phase separation (EPS) in tricolor LaMnO₃/CaMnO₃/PrMnO₃ superlattice grown on SrTiO₃ substrates. The authors obtained the EPS nano-network by thermal cycling the sample through the cubic-tetragonal phase transition in SrTiO₃, and confirmed the structure of such nano ridges by scanning transmission electron microscopy (STEM). Using field-dependent magnetic-force microscopy (MFM), they demonstrated that the EPS network can be easily driven to ferromagnetic (FM) states, in sharp contrast to the robust antiferromagnetic (AFM) bulk. Using scanning microwave impedance microscopy (sMIM), they showed that the EPS network is more conductive than the background domains. Finally, they put forth a density-functional-theory (DFT) based analysis to explain the emergence of FM metallic phase at the nano-network. This work is of high quality. I can recommend its publication in Nature Communications after the authors consider the following comments.

We thank the reviewer for his/her nice summary of our work. The reviewer's comments have been carefully addressed as below.

1. The EPS network was formed during the first cooldown of the sample through the structural transition of SrTiO₃ due to lattice strain at the DWs. The authors claimed that the EPS reappears at the same locations after each round of temperature cycle. However, the SrTiO₃ substrate could still form different tetragonal domain structures at different cooldowns. Why wouldn't they observe more networks during subsequent thermal cycles?

In the revised manuscript, we refrain from stating a direct spatial correlation between the STO domain wall (DW) and the nano-network in superlattice film due to the lack of a direct experimental evidence. Instead, we emphasize the strain relaxation mechanism for the formation of such nano-network as we argued in the original manuscript that “the formation of the nano ridges is likely the consequence of an internal strain relaxation of the tricolor superlattice triggered by the STO DW”. To support it, we provide additional STEM evidence for the existence of structural dislocations at the nano-network regions as the way to release such internal strains (Fig. S3 in SI B). We agree with the reviewer that new DW would form on the STO substrate at different thermal cycles. However, whether the superlattice thin film forms more networks in subsequent thermal cycles depends on if there is additional internal strain that need a further relaxation. It turns out that, it won't happen in two consecutive thermal cycles, i.e., our MFM and sMIM low temperature measurements were conducted in two consecutive cool-downs. This implies that most of the internal strains are released

in the first cool-down. The sample is thus in a structurally relaxed and stable state without the need for further structural relaxation.

2. The MFM data in Fig. 2 clearly demonstrate the FM nature of the EPS network. In contrast, the discussion on sMIM data is problematic. What does the technique measure? How does one relate its signals (in mV from the images) to local conductivity, as claimed by the authors? Such critical information is not found in the text, Methods, or SI.

We thank the reviewer for pointing this out. We have added a detailed introduction to the working principle of scanning microwave impedance microscopy (sMIM) in the supplementary (SI C). We also added one more reference (now reference 20) of sMIM in the revised manuscript. sMIM can be taken as a near-field scanning optical microscopy working at the microwave frequency (3 GHz in our case). By measuring the reflected microwave signal from the sample surface, it measures the tip-sample's microwave impedance which characterizes the screening properties of local sample area. After a proper finite element analysis simulation of the tip-sample's admittance (the inverse of impedance) as shown in the supplementary, one can further show that the measured sMIM signal has a monotonic dependence on the sample's local conductivity, which forms the basis of nano-scale conductivity imaging of sMIM.

3. Following point #2, the sMIM data analysis is not convincing. The authors show that (in Fig. 3b) the 0/90 walls exhibit signals twice as high as the 45/135 walls, without giving the error bars. As far as I can tell, the 45/135 DWs look just the same as the 0/90 DWs, especially in the upper-right and upper-left corners of the images. I also don't find any discussions on why there should be difference in conductivity between these two types of walls. In my opinion, unless they have compelling data and good understanding on the conduction mechanism between these two walls, they should just report the observation of metallic states at the EPS network and the (slight but discernible) field dependence within some experimental errors.

We have added the error bars to the data point in Fig. 3(d) to indicate the spatial variation of sMIM signal levels on DWs for each case. The sMIM signal difference between 0/90 and 45/135 DWs can still be discerned in such a measurement including the error bars, i.e., the averaged conductivity of 0/90 DWs is larger than that of 45/135 DWs. Actually, we have a good understanding of why it is the case. The conductivity of DW is mainly contributed by the ferromagnetic (FM) metallic phases. As stated in the original manuscript, "the higher conductivity in the 0/90 DWs reflects their more uniform ferromagnetism". Our MFM imaging has explicitly demonstrates that a spatially uniform FM phase exists in the 0/90 DWs while it coexists with AFM phase in the 45/135 DWs (see Fig. 2(j) as an example). In other words, the volume fraction of the FM phase of the 0/90 DWs is higher than that of the 45/135 DWs, leading to a higher conductivity in the former case. We agree with the reviewer's observation that the relative sMIM signal of the 45/135 vs. the 0/90 DWs varies spatially. We believe such spatial variation is beyond measurements uncertainties because they correspond well with MFM images, e.g., Fig. 2(j)-(l), where a close comparison between MFM and sMIM images reveals a one-to-one correspondence between the conductivity and the FM phase volume fraction. We have added some discussions on the conductivity variation among 45/135 DWs and its relation to the FM phase volume fraction in the revised manuscript.

4. Fig. 3c is rather out of place. It should go with Fig. 2 instead.

We have replaced the figures as suggested.

Reviewer #2 (Remarks to the Author):

In this manuscript, Li and coauthors present results on a study of manganite superlattices (SLs) (so-called “tri-color superlattices – with lanthanum manganite, praseodymium manganite, and calcium manganite) fabricated on SrTiO₃ (STO) substrates. When the samples are cooled below 105 K and then heated back up, a peculiar domain structure appears in the system, which is apparently caused by the formation of structural twins in the STO substrate below its cubic – tetragonal phase transition which occurs at 105 K. What is rather interesting is that the domain structure then persists in the SL, such that the pattern created is visible by scanning probe microscopy techniques, in both the surface topography, and the local conductivity and magnetic properties. The authors suggest that this is a form of ‘electronic phase separation’ in the manganite layers induced by the local strain from the substrate when it passes the structural phase transition. Various types of domain walls are observed by scanning transmission electron microscopy (STEM), where a significant local strain is identified. Density functional theory (DFT) of the manganite system is then considered to try to understand the influence of the local strain in the domain walls and thus why a local ferromagnetic conductive phase appears within the nominally insulating antiferromagnetic matrix of the system.

The results are interesting, the sample quality appears to be high, and the paper is well presented (aside from some issues with figure presentation; see below) and written carefully with quite good flow. Although I do not recommend publication of this manuscript in the current form, if the authors are able to strongly reply to the below comments and concerns, I feel the paper would be worthy of consideration for publication in Nature Communications.

We thank the reviewer for his/her very comprehensive assessment of our work and the acknowledgement of the quality of this paper. We address his/her comments and concerns as below.

Local strain field and domain wall formation mechanism

On line 69, the authors write, “This nanoscale network strictly follows the structural domain wall (DW) pattern of the underlying STO substrate which exerts a local strain field on the LMO/PMO/CMO superlattice.” But the authors do not show evidence for this local strain field. It would be nice to see direct evidence of the patterns formed in the STO substrate – is there a way to see this in real space on a bare substrate using AFM at low temperature?

We have done low temperature AFM scan on bare STO substrate as suggested by the reviewer, but could not resolve its DW pattern directly (see Fig. R1 below). Although the cubic to tetragonal structural phase transition of STO has been well studied in literatures, experimental investigations mostly rely on diffraction techniques with very few real-space direct imaging (see T.A.Merz *et al.* APL, **108**, 182901 (2016) which uses X-ray micro-Laue diffraction to provide local domain imaging

with $\sim 5 \mu\text{m}$ spatial resolution). Early X-ray diffraction determined the tetragonal lattice distortion at low temperature to be $c/a \sim 1.00056$ (ref. Farrel W. Lytle J. Appl. Phys. **35**, 2212 (1964)). Given the STO cubic phase lattice constant $a = 3.905 \text{ \AA}$, the lattice displacement at the transition is around 0.002 \AA . Such a tiny structural distortion is far below the spatial resolution of our AFM system which explains the absence of STO domain structure at low temperatures in our AFM measurement.

However, we acknowledge that our original statement “This nanoscale network strictly follows the structural domain wall (DW) pattern of the underlying STO substrate which exerts a local strain field on the LMO/PMO/CMO superlattice.” is too strong without direct experimental evidences. This statement has now been modified to “This nanoscale network is a consequence of an internal strain relaxation triggered at the structural domain wall (DW) of the underlying STO substrate at low temperatures.” This statement emphasizes the relationship between the nano-network formation and the strain relaxation, which is supported by our additional STEM measurement to show the evidence of structural dislocations at the nano-network regions as the way to release the internal strain (Fig. S3 in SI B). It also explains why such nano-network pattern persists under subsequent thermal cycles because the sample is already in a strain relieved and structurally stable state. We modified the relevant discussions in the revised manuscript accordingly.

Fig. R1 The AFM measurements on bare STO substrate at (a) 200 K and (b) 2 K, respectively.

Why do the authors run the MFM at 12 K and then the transport measurements by sMIM at 2 K? Wouldn't it be better to do direct comparisons at the same temperatures? Also, why is conductive tip AFM not used to characterize the local transport? It could be a much easier way to also see the conductivity of these local domain walls.

We were simply limited by the lowest temperature one can achieve in our low temperature MFM and sMIM systems which are based on AttoDry 1000 system and AttoDry 2100 system from attocube, respectively. However, the global magnetic and transport properties of the superlattice sample are nearly unchanged from 12 K to 2 K. The transport behavior can be seen from the R-T measurement in the supplementary (Fig. S2 in SI A) indicating that the sample remains in a highly insulating state at low temperatures. To characterize the global magnetic properties, we show magnetization curves M-H taken at 2 K and 12 K (see Fig. R2 below) which also looks similar. Moreover, we switch the scanning head to conduct MFM measurements at 2 K and 12 K in AttoDry 2100 system. The MFM images taken at the same area look almost identical (see Fig. R2 below).

Fig. R2 (a) The magnetization curves taken at 2 K and 12 K. (b) The MFM measurements at 2 K and 12 K. Scale bar is 2 μm .

Although the conducting AFM is a nice probe to perform a local transport measurement, it is not suitable and also challenging to apply it to measure such conductive DWs embedded in an insulating matrix. For a conducting AFM to work, a conductive sample is often required to establish a current path from the tip position to the contact electrode, which is not the case in our tricolor superlattice system. One can tell from the global transport measurements (Fig. S2 in SI A) that the superlattice system is in a highly insulating state indicating that those conductive DWs do not form a percolative network globally. In this regard, sMIM is advantageous for local conductivity imaging. As an AC measurement technique, no contact electrode is needed in sMIM experiment and the tip does not need to be in a direct contact with the sample. Compared to other near-field scanning optical microscopy at infrared frequency, sMIM works at microwave frequency ($\sim\mu\text{eV}$ energy scale). So the AC conductivity determined here faithfully reflects that of DC limits. sMIM has been demonstrated as an ideal probe to study transport behavior in less conductive semiconductor systems (see ref. Mark E. Barber *et al.* Nat. Rev. Phys. **4**, 61-74 (2022)).

What is the mechanism for the domain wall formation and then persistence (even above 105 K when STO becomes cubic again)? It is surprising that the domain structures would appear in the exact same location after several temperature cycles. One would expect that in STO the domains would simply appear at random locations, and these would not be repeatable. More importantly, why do the domains persist in the film layer itself? This is an important point to elucidate because without this we cannot be sure that the effects are not from some localized defects in the samples. Can we see the STEM of the samples near the substrate? Are there any misfit dislocations?

As acknowledged in our above response to the reviewer's question, we don't have direct experimental evidence of the spatial one-to-one correspondence between the STO DW and the observed nano-network in the superlattice thin film, although their pattern shares the same lattice symmetry (Fig. 1b and c). In the revised manuscript, we modified our statement to attribute the formation of nano-network in the superlattice film to an internal strain relaxation process triggered by the structural DW of the underlying STO substrate. This argument gains support from our focus series STEM-HAADF measurements with optical sectioning to show the evidence of structural

dislocations at the nano-network regions as the way to release the internal strain which is also inquired by the reviewer (Fig. S3 in SI B). We agree with the reviewer that in subsequent cool downs, new and different DW pattern will appear in STO. However, the film doesn't have to form new network if there is no additional internal strain to be released. We suspect that most of the internal strains are released in the first cool-down. After that, the sample is in a strain relieved and structurally stable state that doesn't respond to subsequent STO structural DW formation.

Another related question is to do with the repeatability of this experiment. It would be nice to include in the supplementary some other AFM and MFM images of other samples where this phenomenon is observed but the domain structure is different. In other samples, is the same repeatability of the spatial locations of the domain under repeated temperature cycling also shown?

We provide additional AFM, MFM and sMIM images taken on another superlattice sample in the supplementary as suggested by the reviewer (SI D). This new dataset not only demonstrates the repeatability of our experiment in the sense that ferromagnetic metallic phase exists at the nano-network regions forming an electronic phase separation (EPS), but also confirms the persistence of such EPS pattern against the thermal cycle, i.e., the same nano-network reappears at the same spatial locations after a temperature cycle.

On a similar note, it would be very interesting to see that if these SLs are grown on a different substrate (e.g. LSAT) which doesn't exhibit the structural phase transition at 105 K, then upon temperature cycling do these domain structures NOT appear?

We agree. The same superlattice thin film has been grown on NdGaO₃ substrate which doesn't have any structural phase transition at low temperatures. Accordingly, the film stays in a uniform AFM insulating state without any domain structure. These results have been reported in our previous work (see ref. T. Miao *et al.* PNAS, **117**(13), 7090-7094 (2020)). Some MFM images are reproduced and put in the supplementary (SI E) as supporting materials.

Structural and chemical characterization of the samples

More structural and chemical characterization of the superlattices is required. I'm particularly interested in knowing how perfect the claimed single unit cell for each of the three components. Some EDS mapping using STEM here would be very helpful. The RHEED data in Fig S1 does assist but it should be better presented and explained to convince the reader that the samples are indeed what the authors say they are.

The LaMnO₃/PrMnO₃/CaMnO₃ superlattice structure of our film is best evidenced by the X-ray diffraction measurement showing the existence of all superlattice peaks (Fig. S4a in SI B). As suggested by the reviewer, we also perform STEM to characterize the atomic structure of superlattice. Figure S4b in SI B shows the cross-sectional view (along the [110] zone axis) using HAADF-STEM imaging. The intensity line profile along the growth direction clearly resolves the alternating growth of the Pr, La and Ca layers, according to the z-contrast mechanism of HAADF imaging, thus conforms the mono-atomically stacked superlattice structure.

EDS mapping should also be used on the domain walls – what is the chemistry there? Is it still the same as the matrix surrounding or are they a different phase locally?

The chemical composition of the domain walls (in-plane view) was analyzed using STEM electron energy loss spectroscopy (STEM-EELS). By comparing the integrated EELS spectra from domain walls regions and the surrounded domain matrix, no significant elemental difference could be found within the domain wall respect to the surrounded area (Fig. S5 in SI B).

Further structural characterization regarding the average strain state of the films is also required. XRD reciprocal space mapping (RSM) would be nice to confirm that the films are tensile strained on the STO substrate and would be a good way to cross check the results of local mapping using the STEM. It would also be nice to compare the magnitude of the strain change required to induce the transition between AFM and FM states.

We have done a careful X-ray diffraction (XRD) measurements to characterize the strain state of the superlattice film (Fig. S1c and d in SI A). The XRD reciprocal space map as required by the reviewer clearly demonstrates the epitaxial growth of superlattice film on the STO substrate. We further determine the lattice constant of the superlattice film by XRD which gives rise to $a = 3.902\text{\AA}$, $b = 3.904\text{\AA}$ and $c = 3.772\text{\AA}$. Given the STO cubic lattice constant 3.905\AA , the superlattice film is tensile strained on the STO substrate. This is in agreement with the STEM strain analysis in Fig. 1d. It shows a 1.8% reduced lattice spacing at the nano-network regions that answers the reviewer's question of the magnitude of the strain change needed to induce the transition between AFM to FM states.

Figures and presentation of the data

A small but important point. This is less critical for the main figures as they should be copyedited before publication, but for the supplementary information, the text size in the figures is much too small to be legible. This is particularly problematic for Fig. S1 b,d, and Fig. S2. But even better would be to make font size uniform throughout the figures to improve readability. This latter point applies to all the figures, including in the main text. An extreme case is the labels in Figure 1a. These are not even readable. The authors must go through all figures and carefully check the font type (pls make all uniform, e.g. Arial) and sizes.

Figure S1 is too small to read, especially the RHEED and XRD data. Please either make larger or reconsider the layout (could separate into two figures). Also, please index the XRD peaks in Figure S1d.

Figure 4a – please include a bigger schematic and put the colors of the various atoms.

Thanks for pointing it out. We have taken care of them in the revised manuscript to make sure all the figure presentations are easily readable and consistent.

More exciting experiments and work towards applications

It would be highly desirable to show manipulation of the nanoscale EPS network, either by applied force through the AFM tip, or maybe some kind of substrate manipulation (pre-etching and/or in-situ bending?). This kind of manipulation would significantly increase the impact of this paper. I encourage the authors to consider this, and if possible, include such data (even if preliminary) in the supporting information.

We agree with the reviewer that showing the possibility of active manipulation of the nano-network EPS state in our film is essential for the significance of this work. We take advices from the reviewer to use AFM tip to locally apply a force to the film. Interestingly, a narrow ferromagnetic metallic stripe was generated after such a tip mechanical perturbation. We added this preliminary result in the supplementary as suggested by the reviewer (SI G). This strongly indicates that the tricolor superlattice film is highly sensitive to the local strain perturbation and thus subjected to an active manipulation.

Others

The authors should include some other references regarding changes in chemistry and structure at domain walls. One reference that is relevant is Farokhipoor et al., “Artificial chemical and magnetic structure at the domain walls of an epitaxial oxide,” Nature 2014.

We have added this paper as a reference (now reference 13) when we discuss the chemistry and structure of our nano-network in the revised manuscript.

In the abstract (line 28), the authors write “controlled formation of EPS nanoscale network...” – this is probably an oversell since the network appears to be formed in a random nature through stochastic processes during the STO structural phase transition. I suggest the authors tone this down.

We accept the reviewer’s suggestion and removed the “controlled” in the abstract.

Reviewer #3 (Remarks to the Author):

This paper reports on nanoscale electronic phase separation (EPS) that emerges at the twin domain boundaries formed in epitaxial thin films of perovskite manganites. The presence of EPS was directly identified by employing local probes for magnetic (MFM) and conducting (MIM) states. Based on the results of these real-space observations, the authors have concluded that the strain relaxation at the domain boundaries induces the FM-metallic state under magnetic fields while the other strained regions are kept to the A-type AFM-insulating state. The conclusion is reasonable and supported well by experimental results. In particular, there are very few reports of direct observations of local EPS by both magnetic and conductive probes at the same positions, which are worth publishing somewhere.

However, many papers have already reported the dramatic change in the electronic state via epitaxial strain in manganite thin films. It is also known that biaxial tensile strain tends to stabilize the A-type AFM state, as reported by Z. Fang et al., Phys. Rev. Lett. 84, 3169 (2000). There are also many papers reporting the real-space observations of nanoscale EPS in manganite films employing various local probes. Even the observation of EPS induced by local strain has already been reported in A. S. McLeod et al., Nat. Mater. 19, 397 (2020). What is distinctly new about this study compared

to previous papers is the finding of EPS developing only at the twin domain boundaries. However, the domain boundaries are unintentionally introduced associated with the structural phase transition in the substrates, and hence, their positions are uncontrollable. The origin of the FM-metallic state at the domain boundaries is the epitaxial strain relaxation, which is a known mechanism. Therefore, I feel that the present work lacks exciting novelty to provide substantial advances in this research field and cannot recommend the publication in Nature Communications. The followings are comments on more detailed points.

1. It is still unclear why the tricolor superlattice is necessary for this work. The sample is nominally a superlattice consisting of mono-atomic layers of three compounds, but the electronic structure will be almost the same as that of the mixed compound in such a short-period superlattice. Even the chemical structure is seemingly very close to the mixed one, judging from the absence of the contrast along the stacking direction associated with the difference in the A-site elements in the cross-sectional STEM images shown in Figs. 1(d) and 1(e), probably due to the chemical diffusion and interface roughness. It is necessary to explain in more detail the advantage of the superlattice structure in this work.

We thank the reviewer for the nice summary of our work and the acknowledgement of our dual-probe approach to the study of EPS in manganite. His/her concerns mainly lie in the novelty of this work in comparison to previous ones for which we carefully explain as below. It is also closely related to the reviewer's first question that why we use tricolor superlattice in this work.

Before we get to the novelty issue, we first clarify the STEM results shown in Fig. 1(d) and (e) which are in-plane view STEM-HAADF images, rather than the cross-sectional ones. Therefore, one should not expect to see any superlattice A-site contrast. We show the cross-sectional view STEM-HAADF images in the supplementary (Fig. S4 in SI B) to display a clear A-site ordered superlattice structure of our sample.

Regarding why using the tricolor superlattice to study EPS, we point out that the choice of such sample shows exactly the novelty of our work. While many papers (including several from our own group) have reported real-space observations of EPS states in manganites using various scanning probes (MFM, SNOM or sMIM), they are completely different from the EPS nano-network in this work. This is because samples used in previous works have a tendency to be in a global EPS state featuring domains with somewhat irregular shapes and intertwined configurations. In stark contrast, the present superlattice sample is in a robust antiferromagnetic insulating state, which can only be locally turned into ferromagnetic metallic state by strain relaxation with all other regions remained in antiferromagnetic insulating state. This provides a platform for a possible controlled patterning of EPS state using local strain field. Some preliminary results have been shown in the supplementary (SI G) in which a tip mechanical pressing generates a local ferromagnetic metallic phase on the superlattice film.

This contrasting behavior can be better appreciated by comparing this work to one previous work from our group (see Y. Zhu *et al.* Nat. Commun **7**,11260 (2016)), in which we grew two kinds of $(\text{La}_{1-y}\text{Pr}_y)_{1-x}\text{Ca}_x\text{MnO}_3$ (LPCMO) thin films on STO substrate. One has a random A-site mixing, and the other has a partial A-site order with $[\text{La}_{0.625}\text{Ca}_{0.375}\text{MnO}_3/\text{Pr}_{0.625}\text{Ca}_{0.375}\text{MnO}_3]$ superlattice. However, both LPCMO films show conventional EPS state with spatially intertwined configurations even they experience the same strain field from STO as the $\text{LaMnO}_3/\text{CaMnO}_3/\text{PrMnO}_3$ superlattice

(see images below). It is apparently the formation of a fully A-site ordered LPCMO film that gives rise to such a peculiar local strain response and a new type of EPS state in manganites.

Fig. R3 MFM images of LPCMO with random A-site mixing (a) and [LCMO/PCMO] superlattice with a partial A-site order (b). These images are reproduced from the reference Y. Zhu *et al.* Nat. Commun **7**,11260 (2016). Scale bar is 2 μm .

The reviewer mentions the work of A. S. McLeod *et al.*, Nat. Mater. **19**, 397 (2020), which is an important reference to our work as they both employ a dual-probe approach. McLeod's work also reported a ferromagnetic metallic (FMM) phase forming in a crack on the surface and attributed it to the strain relaxation. However, the scientific object differs between these two works. In McLeod's work, they chose $\text{La}_{2/3}\text{Ca}_{1/3}\text{MnO}_3$ system which doesn't feature EPS. The authors mainly focused on a photo-induced "hidden" FMM phase and studied its dynamic growth and melting process. Our focus is to study EPS of manganites and its manipulation. So we intentionally chose LPCMO system featuring a pronounced large length-scale EPS and reported a local strain engineering of EPS state in $\text{LaMnO}_3/\text{CaMnO}_3/\text{PrMnO}_3$ superlattice film.

On the theory side, we agree that the strain effect (A-AFM order stabilized by tensile strain) discussed in early work of Z. Fang *et al.*, Phys. Rev. Lett. **84**, 3169 (2000) holds true under many circumstances in manganites. However, our first principles calculation surely has its merits as it specifically studies the $\text{LaMnO}_3/\text{PrMnO}_3/\text{CaMnO}_3$ tricolor superlattice system which doesn't exist before. The fact that the new tricolor superlattice shows a similar trend to be A-AFM under a tensile strain only suggests the robustness of the underlying physical process among different manganite systems. The same argument applies to the work of A. S. McLeod *et al.*, Nat. Mater. **19**, 397 (2020) as well in which they conduct a density function theory calculation and reach a similar conclusion.

We have made a better clarification of the novelty of our work and the usage of the superlattice film in the revised manuscript.

2. The lattice constant of STO is generally 3.905 Å at room temperature. However, the in-plane lattice constant of the film in the strained region evaluated from the STEM image (Fig. 1(d)) is much larger (about 3.96 Å). Why is the in-plane lattice constant of the film so large? Moreover, no results have been shown to prove that the in-plane lattice of the thin film is locked to the substrate.

We thank the reviewer for pointing this out. We have carried out a careful X-ray diffraction (XRD) measurement to determine the lattice constant and the strain state of our superlattice film grown on STO substrate (Fig. S1c and d in SIA). The XRD results unambiguously demonstrate that the superlattice film is tensile strained on the STO substrate, i.e., the in-plane lattice constant of the film is around 3.90\AA which is locked to the STO substrate.

Regarding the STEM result of 3.96\AA , we note that the absolute value of lattice spacing measured by STEM is not very accurate. This is limited by the intrinsic imaging mechanism of STEM that the magnification is determined by the scanning step sizes of the scanning coils. The displayed lattice distance is subjected to the calibration of the microscope, which is known for STEM measurements (see ref. Brydson, Rik, ed. Aberration-corrected analytical transmission electron microscopy. Vol. 280. Chichester: Wiley, 2011.). However, the strain measurements based on the STEM images (Fig. 1d and e) are accurate for which only relative changes of the lattice spacing need to be measured. On the other hand, XRD provides more accurate crystallographic structure information than TEM. The results only rely on the x-ray wavelength and the geometric distance from the sample to the detector.

To avoid the potential controversy, we change the line-profile of Fig. 1d and e to the relative change of the averaged lattice spacing (in percentage unit) compared to the domains at two sides in the revised manuscript.

3. The authors attribute the reduction of the resistance under magnetic fields shown in Fig. S2(a) to the metallic phase at domain boundaries. More detailed consideration will be necessary about whether the change in the conduction at the domain boundaries can explain the observed magnetoresistance.

We think the reviewer may have some misunderstanding of this part of the paper. When we discuss the R-T in Fig. S2a, we state in the original supplementary “Figure S2a displays the resistance versus temperature at different magnetic fields. The film is in an insulating state with almost no thermal hysteresis up to 9 T, indicating the absence of electronic phase separation in such a tricolor superlattice similar to previous reports”. We didn’t mention the reduction of the resistance under magnetic fields and relate it to the conductive nano-network. Actually, one can only tell an insulating state from such a global transport measurement. The existence of metallic nano-network was later revealed by our real-space MFM and sMIM probes. The fact that the system manifests as an insulating state in the global transport suggests that such metallic domain boundary doesn’t actually form a percolating conductive network throughout the sample.

Below is a list of the main changes to the manuscript:

- (1) We lowered our tone in various places of the revised manuscript concerning the relationship between STO domain walls (DWs) and nano-network seen in superlattice film. We don’t state their one-to-one spatial correspondence due to the lack of a direct experimental evidence. Instead, we emphasize that the formation of nano-network is a consequence of an internal strain

relaxation triggered by STO DWs.

- (2) We made a better explanation to the novelty of our work at the end of the manuscript, i.e., the usage of a fully A-site order LPCMO superlattice film ensures such a local strain engineering and EPS state with an excellent spatial selectivity.
- (3) We made substantial change to the supplementary materials by adding new experimental results from XRD, TEM and scanning probe (AFM, MFM and sMIM) to address the reviewer's questions regarding the working principle of sMIM, the structure and chemistry of the superlattice and nano-network, the repeatability of our observation and the potential manipulation of such nano-network EPS state.
- (4) We modified the figure display both in the main text and the supplementary to accommodate the change of the text.
- (5) We added two more references (ref. 13 and 20).
- (6) We added two more authors who made contributions to this revised manuscript.

Prepared by:

Jian Shen and Xiaodong Zhou

REVIEWER COMMENTS

Reviewer #1 (Remarks to the Author):

In the revised manuscript, the authors have carefully addressed my comments and suggestions. I can now recommend its publication in Nature Communications.

Reviewer #2 (Remarks to the Author):

In their revised manuscript and rebuttal letter, the authors have carefully considered the comments from the Referees. In my view, they have taken all the comments seriously, and have taken the effort and the time to perform additional experiments and to explain their reasoning carefully.

My key issues regarding repeatability of the experiments, explanation of the strain field from STO substrate, and more detailed structural analysis have been addressed. The revisions made to the manuscript make the work a lot more robust. I thank the authors for their work and detailed discussions.

I am pleased to recommend publication, after consideration of the following very minor points.

Figures S1 and S4 – please report the wavelength of the x-rays. There is some inconsistency with the position of the peaks in the XRD data between these two figures.

Figure S4 is too small – the text and features of XRD and STEM are difficult to discern.

Figure S9 – what is the meaning of the red circles in this figure?

Reviewer #3 (Remarks to the Author):

I would like to thank the authors for their thoughtful responses to the reviewers' comments and questions. The revised manuscript clarifies the importance of employing the tricolor superlattice and the significance of the results in this paper. Therefore, I recommend the publication of this paper in Nature Communications.

In this letter we provide a point-to-point response to the reviewers' comments.

In the following, the reviewer's original comments are shown by blue italic characters.

The authors' responses are shown by black normal characters.

Reviewer #1 (Remarks to the Author):

In the revised manuscript, the authors have carefully addressed my comments and suggestions. I can now recommend its publication in Nature Communications.

We thank the reviewer for reviewing our paper and his/her recommendation for its publication.

Reviewer #2 (Remarks to the Author):

In their revised manuscript and rebuttal letter, the authors have carefully considered the comments from the Referees. In my view, they have taken all the comments seriously, and have taken the effort and the time to perform additional experiments and to explain their reasoning carefully.

My key issues regarding repeatability of the experiments, explanation of the strain field from STO substrate, and more detailed structural analysis have been addressed. The revisions made to the manuscript make the work a lot more robust. I thank the authors for their work and detailed discussions.

I am pleased to recommend publication, after consideration of the following very minor points.

We thank the reviewer for reviewing our paper and his/her recommendation for its publication. We address his/her comments as below.

Figures S1 and S4 – please report the wavelength of the x-rays. There is some inconsistency with the position of the peaks in the XRD data between these two figures.

The X-ray reciprocal space maps and the XRD measurements in Fig. S1 were collected by using Cu-K α radiation ($\lambda = 1.5418 \text{ \AA}$, Bruker AXS D8 Discover). Synchrotron XRD measurements in Fig. S4 were carried out at beamline BL14B1 of Shanghai Synchrotron Radiation Facility at room temperature, using 10 keV X-rays ($\lambda = 1.239 \text{ \AA}$). Therefore, the angle position of the same peak in the XRD data in Fig. S4 is smaller than that of Fig. S1 as pointed out by the reviewer. We have provided this information in the method section of the revised manuscript.

Figure S4 is too small – the text and features of XRD and STEM are difficult to discern.

We enlarge Fig. S4 so it can be better viewed.

Figure S9 – what is the meaning of the red circles in this figure?

Sorry for missing this information. The red circles denote the defects in the scanned area of Fig. S9. It served as a marker to make sure the same sample area was scanned at different magnetic fields for such almost featureless MFM imaging. We added this information in the revised supplementary.

Reviewer #3 (Remarks to the Author):

I would like to thank the authors for their thoughtful responses to the reviewers' comments and questions. The revised manuscript clarifies the importance of employing the tricolor superlattice and the significance of the results in this paper. Therefore, I recommend the publication of this paper in Nature Communications.

We thank the reviewer for reviewing our paper and his/her recommendation for its publication.

Below is a list of the main changes to the manuscript:

- (1) We modified the method section of the main text to provide the wavelength of the x-rays used in XRD measurements.
- (2) We modified the data description of Fig. S9 in the revised supplementary regarding the defect areas denoted by red circles.

Prepared by:

Jian Shen and Xiaodong Zhou